# Ego4D Goal-Step: Toward Hierarchical Understanding of Procedural Activities

**Yale Song**    **Eugene Byrne**    **Tushar Nagarajan**
**Huiyu Wang**    **Miguel Martin**    **Lorenzo Torresani**
Fundamental AI Research (FAIR), Meta
https://github.com/facebookresearch/ego4d-goalstep

## Abstract

Human activities are goal-oriented and hierarchical, comprising primary goals at the top level, sequences of steps and substeps in the middle, and atomic actions at the lowest level. Recognizing human activities thus requires relating atomic actions and steps to their functional objectives (what the actions contribute to) and modeling their sequential and hierarchical dependencies towards achieving the goals. Current activity recognition research has primarily focused on only the lowest levels of this hierarchy, i.e., atomic or low-level actions, often in trimmed videos with annotations spanning only a few seconds. In this work, we introduce Ego4D Goal-Step, a new set of annotations on the recently released Ego4D with a novel hierarchical taxonomy of goal-oriented activity labels. It provides dense annotations for 48K procedural step segments (430 hours) and high-level goal annotations for 2,807 hours of Ego4D videos. Compared to existing procedural video datasets, it is substantially larger in size, contains hierarchical action labels (goals - steps - substeps), and provides goal-oriented auxiliary information including natural language summary description, step completion status, and step-to-goal relevance information. We take a data-driven approach to build our taxonomy, resulting in dense step annotations that do not suffer from poor label-data alignment issues resulting from a taxonomy defined a priori. Through comprehensive evaluations and analyses, we demonstrate how Ego4D Goal-Step supports exploring various questions in procedural activity understanding, including goal inference, step prediction, hierarchical relation learning, and long-term temporal modeling.

## 1 Introduction

Recognizing complex patterns of human activities has been the subject of extensive research in computer vision and the broader AI community [2, 25, 46, 8, 23, 22]. However, progress has been relatively slow compared to object and scene understanding [59, 32, 36, 62]. One of the main obstacles has been the scarcity of large-scale datasets annotated with a comprehensive taxonomy representing complex human activities. While object recognition benefits from WordNet [15] that provides an extensive taxonomy of objects found in everyday scenarios, activity recognition is presented with unique difficulties because there is currently no established taxonomy in place that encompasses the broadly varying granularities of activities, from atomic actions (e.g., pick-up cup, sit down) to procedural sequences (e.g., make lasagna).

In our quest to build a new dataset for human activity recognition, we draw inspiration from the psychology literature. Studies have shown the inherent *hierarchical* nature of human behavior [5, 12], comprising the primary **goals** at the highest level, intermediate **steps** and their **substeps** in the middle, and **atomic actions** at the lowest level. Social cognitive theories [3] suggest that this hierarchy is formed by human agents deliberately setting goals, anticipating potential consequences of different

37th Conference on Neural Information Processing Systems (NeurIPS 2023) Track on Datasets and Benchmarks.

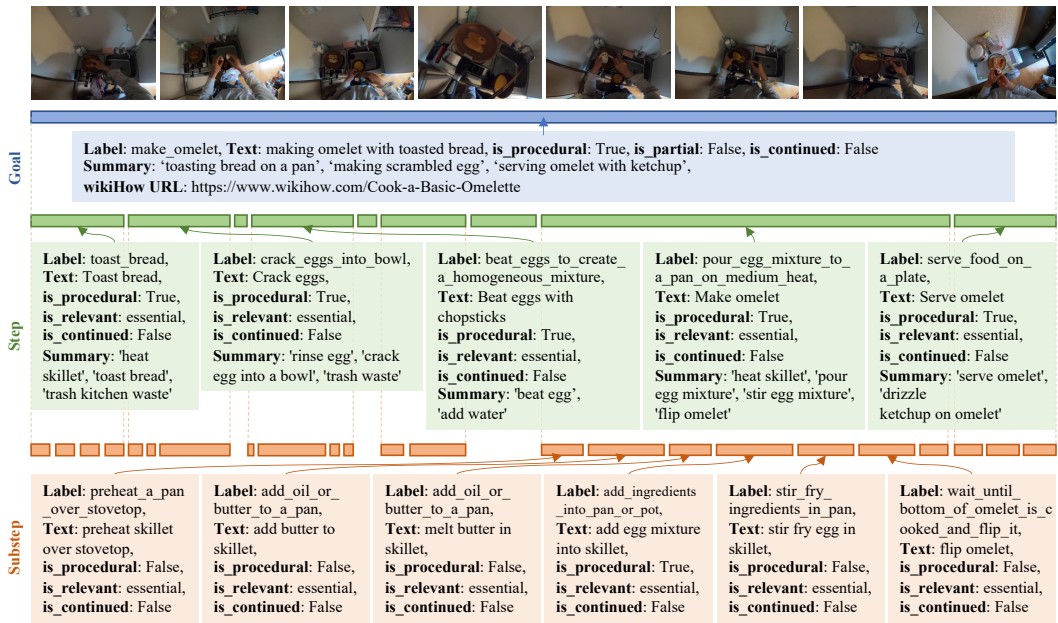

Figure 1: **Ego4D Goal-Step** offers hierarchical procedural activity labels with three distinct levels: goal - step - substep. Each annotation comes with a time interval, categorical label, and natural language description. It also provides auxiliary information including step summaries, task completion status (is_continued), task relevance (is_relevant), and procedural activity indicator (is_procedural).

actions, and planning out a sequence of steps and their substeps to achieve the desired goal in a hierarchical manner. Although the planned sequence of actions may not necessarily align with the actual execution order [5], inferring and reasoning over the hierarchical representations has been shown to be crucial for understanding human behavior [12, 41].

Most existing activity datasets have focused on the lowest levels of this hierarchy – i.e., atomic or low-level actions – often in trimmed videos and with annotations spanning only a few seconds [25, 46, 8, 23]. This focus on atomic actions has even raised questions about the necessity of temporal modeling in existing video tasks [40, 11, 44], and its suitability for studying real-world videos containing higher-level activities over longer temporal extents.

In response to this, *procedural activities* – those that involve performing a series of *steps* to achieve predetermined *goals* – have recently gained particular attention [29, 60, 49, 61, 48, 34, 4, 43, 55]. Recognizing goal-oriented steps that unfold over a long time horizon requires modeling long-term temporal context, making it a challenging long-form video understanding task. However, existing datasets are either small-scale [29, 3, 43], do not model high-level goals, or ignore the hierarchical relationship between steps [49, 61, 4]. Furthermore, the step taxonomy is commonly built from external sources detached from videos (e.g., text articles from wikiHow [60, 49, 61]), resulting in misalignment between the constructed label space and the observed data. Consequently, a significant portion of the video is left unlabeled, offering an incomplete record of activities.

To address these limitations, we introduce **Ego4D Goal-Step**, a new set of annotations on Ego4D [22] with a newly developed hierarchical taxonomy for procedural activities. It contains two main components: (1) The goal annotation set consists of 7,353 videos, totaling 2,807 hours, labeled with a taxonomy of 319 goals grouped into 34 scenarios. This is a focused set covering the 72% of Ego4D videos and is intended to provide a large-scale training and evaluation dataset for goal inference. (2) The step annotation set focuses on the cooking scenario portion of Ego4D, and is intended specifically for procedural activity recognition. It consists of 47,721 densely labeled step and substep segments, amounting to 430 hours in total, annotated based on a taxonomy of 86 goals and 514 fine-grained steps commonly performed in diverse home kitchen environments across many countries.

Ego4D Goal-Step stands out from existing procedural video datasets for its several unique features. **(1)** We develop our taxonomy in a data-driven manner to accurately represent the activities, rather than

| Dataset | Hier | Video Statistics | | Segment Statistics | | | Taxonomy | |
|---|---|---|---|---|---|---|---|---|
| | | Count (total) | Duration (total / avg) | Count (total / avg) | Duration (total /avg) | Density (Fg ratio) | # Goals | # Steps |
| EgoProceL [4] | | 329 | 62h / 13m | 1K / 8.7 | 24h / 16.3s | 0.38 | 16 | 139 |
| YouCook2 [60] | | 2,000 | 176h / 5m | 15K / 7.7 | 84h / 19.6s | 0.48 | 89 | n/a |
| CrossTask [61] | | 2,750 | 375h / 5m | 21K / 7.6 | 56h / 9.6s | 0.15 | 18 | 133 |
| COIN [49] | | 11,827 | 468h / 2m | 46K / 3.9 | 192h / 14.9s | 0.41 | 180 | 778 |
| Breakfast [29] | ✓ | 2,193 | 85h / 2m | 62K / 28 | 108h / 6.3s | 0.88 | 10 | 225 |
| Assembly101 [43] | ✓ | 362 | 43h / 7m | 83K / 237 | 156h / 1.7s | 0.81 | 101 | 1,582 |
| **Ego4D Goal-Step** | ✓ | 851 | 368h / 26m | 48K / 56 | 430h / 32.5s | 0.77 | 86 | 514 |
| **Ego4D Goal-Step (goal labels)** | | 7,353 | 2,807h / 23m | 7546 / 1 | 2,470h / 19.6m | 0.88 | 319 | - |

Table 1: **Dataset statistics.** We report "Ego4D Goal-Step" — the subset that comes with hierarchical step labels, and "Ego4D Goal-Step (goal labels)" — the full set which includes videos with goal labels but no step labels. "Hier" indicates datasets with hierarchical label spaces. Breakfast [29] and Assembly101 [43] provide both coarse- and fine-level segments, analogous to our step and substep segments; we report their combined numbers. Assembly101 [43] provides 12 synchronized views per recording; we report single view statistics to make the numbers compatible with other datasets.

resorting to external resources such as wikiHow. As a result, the average density of annotated segments per video is 77%, i.e., 2-5× higher than existing datasets that rely on external taxonomies [61, 49]. **(2)** The annotations form a three-level hierarchy (goal - steps - substeps), a unique feature that is unavailable in most existing datasets. When combined with existing Ego4D labels that are action-centric (e.g., narrations, FHO, and moments), it creates an attractive multi-level hierarchical label space. **(3)** Every annotation includes a time interval, a categorical label, and a natural language description, enabling both detection and language grounding tasks. **(4)** Procedural segments also come with step summary descriptions, useful for video summarization [38]. **(5)** Additional goal-oriented information, such as task relevance and completion status signals, supports novel research directions such as task graph inference and progress tracking [33]. **(6)** Our dataset inherits the large scale and diversity of Ego4D, capturing immersive views of procedural activities from a first-person perspective, and featuring long untrimmed videos that reveal the unfolding of individual goals over time. Together, these strengths make Ego4D Goal-Step a significant step forward in procedural activity understanding. See Figure 1 for an illustration of these features.

In summary, we introduce Ego4D Goal-Step, the largest available egocentric dataset for procedural activity understanding, with 2,807 hours of videos with specific goal labels and 430 hours of segments with fine-grained step/substep labels. In what follows, we describe our annotation and taxonomy development process. We also provide a comprehensive analysis of new annotations and compare them with existing procedural video datasets. Finally, we demonstrate the value of our annotations for temporal detection and grounding tasks, and analyze the results in the context of the unique properties of our dataset.

## 2 Related Work

Activity recognition has a two-decade history in computer vision. Early works have tackled atomic action classification using datasets of seconds-long video clips at relatively small scales [42, 20, 35, 30, 47]. Following the success of deep neural networks, several datasets have focused on scaling up by leveraging online videos [27, 1, 8, 28, 21, 37]. Recognizing the need for long-form video modeling, several datasets have also been proposed for action detection in untrimmed videos [6, 46, 26, 54, 23].

Recently, the community has expanded the scope by developing datasets for procedural activities. The typical dataset construction process involves selecting procedural tasks beforehand, e.g., various recipes in cooking, then collecting data for the pre-selected tasks through participant recordings or by mining online video repositories. Participant-recorded datasets like Breakfast [29] and Assembly101 [43] benefit from a controlled collection setting, enabling the development of a taxonomy aligned with the data, and resulting in dense annotations and hierarchical step segments similar to ours. However, they capture limited diversity (e.g., 10 cooking recipes) and are smaller in scale.

On the other hand, Cross-Task [61] and COIN [49] are Internet-mined datasets that benefit from the scalability. However, they rely on external resources to develop a taxonomy (e.g., wikiHow), leading to label spaces that do not capture precisely and comprehensively the activities represented in the videos. Consequently, a large portion of videos remains unlabeled, with densities of labeled segments in the low 40% (see Table 1). Furthermore, annotated segments often only partially match with step labels (e.g., due to subtle variations in objects used, step ordering, etc.), resulting in weakly-labeled data. These datasets are also often non-hierarchical and represent steps at a single level.

Compared to existing procedural video datasets, Ego4D Goal-Step provides dense annotations at scale, a well-aligned taxonomy, and a hierarchical label space – all at the same time – thanks to our hierarchical partitioning [10] approach to data annotation and taxonomy development. With a wide range of procedural videos sourced from Ego4D, accompanied by dense annotations with a comprehensive taxonomy, our dataset enables procedural activity understanding at scale.

## 3 Ego4D Goal-Step

### 3.1 Ego4D: The Status Quo

Ego4D is annotated in a variety of ways. All videos are annotated with scenarios and narrations which provide high-level and low-level descriptions of actions, respectively. Scenarios provide coarse categorization of activities (e.g., construction, arts and crafts), while narrations describe a camera-wearer's action at a specific time. The narrations are interaction-centric, focusing on individual, *atomic* actions that a camera-wearer performs over a short time period. For example "C picks up the spoon," "C pets the dog," "C unscrews the screw," where C represents the camera-wearer.

While the narrations offer valuable information to understand simple movements and hand-object interactions, in the broader context of activity understanding, they are limited. Human actions are not performed arbitrarily — they are *intentional* and are done to accomplish a particular *goal* [3]. For example, C picks up the spoon *to add sugar to coffee*; C unscrews the screw *to detach a bicycle wheel*. These goals are left hidden in existing narrations. Moreover, these goals themselves are part of more structured activities. "Adding sugar" is a step in the process of making coffee, and "Detaching the bicycle wheel" is a step towards replacing a punctured tire tube. While narrations explicitly capture what is immediately happening, they do not reveal *why*, or more broadly, to what end.

Narrations form the scaffolding for various other annotations on smaller subsets of Ego4D. Forecasting hands and objects (FHO) annotations involve atomic actions parsed into simpler (verb, noun) tuples. For example, the long-term forecasting task involves predicting the sequence of future atomic actions, without capturing the overarching goal. Moments query annotations go one step higher, representing composite actions that involve sequences of atomic actions like "wash dishes in the sink" or "put on safety equipment." While they are higher-level, they are still short-term activities and are not connected by their long-term task structure. Moreover, they cover a small set of categories (roughly 100), span an inconsistent set of granularities (e.g., the atomic action "cut dough", and the high level "operate the dough mixing machine"), and are not intended to cover complete activities/goals.

Collectively, these annotations inherit the narrow scope of narrations and offer only a short-term understanding of human activity, limiting their value for procedural activity understanding in intentional, long-form and structured human activity.

### 3.2 Ego4D Goal-Step: Annotation and Taxonomy Development

To address this gap, we annotate Ego4D for procedural video understanding. Ego4D videos are collected without prearranged scripts or step-by-step instructions. As a consequence, the complete set of activities present in the dataset is unknown and a taxonomy cannot be established beforehand. We overcome this in a data-driven manner, using a hierarchical partitioning approach for annotation and taxonomy development. In short, we first ask annotators to identify the primary goals depicted in each video. Next, they delve into each goal segment to identify the individual steps and their corresponding action sequences. Then, again, annotators recursively analyze each action segment to further annotate steps at lower levels to construct a complete step hierarchy. Throughout this process, we present the annotators with an incomplete and evolving taxonomy and encourage them to

suggest missing categories. We review them periodically and update our taxonomy over the course of annotation. The whole process involves five stages:

**Stage 1: Goal taxonomy initialization**    We initialize a goal taxonomy with Ego4D scenario labels and manually subdivide them into specific goal categories. For example, we expand the "`cooking`" scenario into popular dishes (e.g., "`make pasta`," "`make omelet`") and the "`construction site`" scenario into specific construction tasks (e.g., "`paint drywall`," "`build wooden deck`").

**Stage 2: Goal annotation and taxonomy update**    Annotators watch full-length videos (not clips cut to shorter lengths) and identify distinctive goal segments, assigning a goal category and providing free-form text description, e.g., `make_omelet` and "making omelet with toasted bread" as shown in Figure 1. We emphasize the importance of providing full-length videos to the annotators because short-length clips will inherently lack the overall goal of actions (e.g., a clip capturing boiling salted water will not give any clue that it was part of making pasta). For missing categories, annotators choose "`other`" and describe it in text. Auxiliary information, including procedural activity indicator (`is_procedural`), task continuity from previous segments (`is_continued`), and bullet-point summaries of procedural steps, are also annotated.

While the annotation is in progress, we iteratively refine our goal taxonomy. This process is largely manual. It involves mapping keywords in descriptions to existing goal categories and adding new categories to the taxonomy when necessary. We also manually verify the correctness of keyword mapping by visual inspection. This process is done periodically in batches of annotations.

**Stage 3: Step taxonomy initialization**    We prompt a large language model, LLaMA-7B [50], to generate step-by-step instructions for each goal. This provides a concise, but potentially incomplete, list of step candidates per goal category. Moreover, it often misses routine steps performed in a given environment, e.g., washing hands in a home kitchen. To represent missing steps, we create a "catch-all" bucket for each scenario and use the same model to generate commonly occurring steps.

**Stage 4: Step annotation and taxonomy update**    Annotators review full-length videos with goal annotations and identify step segments, assigning a step category and natural language description. To ensure annotators capture step-level granularity that reveals *intentions* and not low-level physical movements, we provide a specific template, *"the camera wearer did X in order to Y"*, and ask them to prioritize intentions (Y) over physical movements (X) in their annotations. As shown in Figure 1, this allows us to collect labels that reveal the functional objectives behind actions (e.g., "toast bread," "crack eggs," "beat eggs with chopsticks") rather than just the description of low-level actions.

Similar to stage 2, annotators indicate the procedural nature of step segments and describe substeps in bullet-points. Additionally, they provide task-relevance information (`is_relevant`) for each step segment using a multiple-choice question ("`essential`," "`optional`," and "`irrelevant`"). They are further requested to find a relevant wikiHow article and provide its URL, for annotation accuracy and educational resources. The taxonomy update follows the same process as stage 2.

**Stage 5: Substep annotation and taxonomy update**    Finally, we ask annotators to further partition step segments into individual substeps. This process is largely the same as stage 4, and it shares the same step taxonomy. The only difference is that we provide annotators with both the corresponding goal and step annotations, and that we encourage annotators to focus on lower levels of granularity.

### 3.3    Dataset Analysis

**Statistics**    Table 1 compares Ego4D Goal-Step with existing procedural activity datasets. As shown, Ego4D Goal-Step represents the largest available procedural video dataset in terms of total number of hours annotated with step labels. Goal annotations are available for a subset covering 72% of Ego4D after discarding non-procedural or uninteresting videos, totaling 2,807 hours of videos. At the segment level, there are a total of 47,721 segments for steps and substeps combined. On average, each goal segment has 23.82 step segments, and each step segment has 4.6 substep segments.

**Annotation**    We contracted a third party vendor to manage the annotation process, and checked privacy and ethical compliance through rigorous reviews. We completed 10 iterations for goal annotation (stage 2), 10 iterations for step annotation (stage 4), and 8 iterations for substep annotation (stage 5). Each iteration involved annotators providing labels for a batch of videos, us reviewing them, and updating the taxonomy. On average, annotators took $2\times$ the video duration to annotate

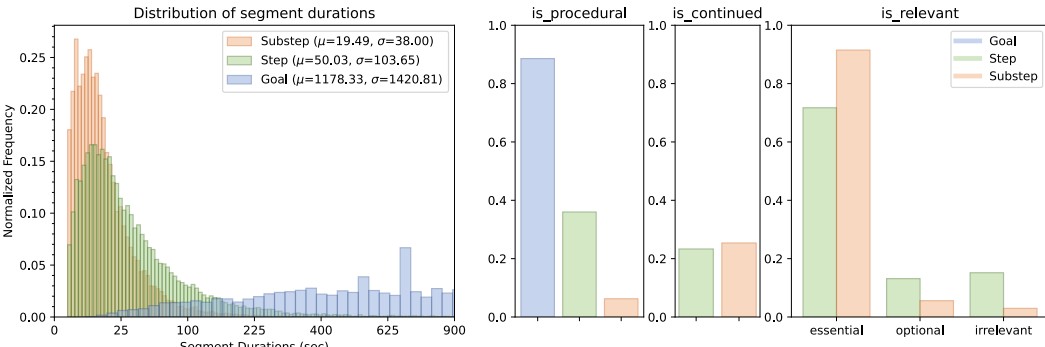

Figure 2: **Dataset statistics** illustrating the distributions of segment durations (x-axis scaled to the square root for better display) and auxiliary labels across different levels of hierarchy.

goals, and 7.5× to annotate steps/substeps because step annotation required fine-grained inspection. The total annotation time amounted to roughly 10,000 worker hours.

**Granularity** Figure 2 illustrates the distribution of goal, step, and substep segment durations. On average, goal segments span 19.64 minutes, while step and substep segments last 50.03 seconds and 19.49 seconds, respectively. The average substep duration aligns with datasets that derive their taxonomy from wikiHow [4, 60, 49], which suggests that our data-driven taxonomy has granularity similar to that found in annotations based on fixed step-by-step instructions. The average duration of our steps/substeps is 32.5 seconds, which is 6 times longer than Breakfast [29] and 22 times longer than Assembly101 [43]. This shows that most of the step/substep segments capture longer-duration actions than [30, 43] without sacrificing annotation density or capturing short-term atomic actions.

**Taxonomy** Our taxonomy contains 319 goal categories grouped into 34 scenarios. Among these, the cooking scenario provides procedural step annotations, comprising 86 goal categories and 514 step categories. We use these step categories from the taxonomy to annotate both step and substep segments. The dataset exhibits a long-tail distribution, with 125 goal categories representing 90% of labeled goal segments and 209 step categories covering 90% of labeled step/substep segments.

**Labels** Ego4D Goal-Step comes with various goal-oriented activity labels. About 92% of the goal segments are annotated with descriptions (3.03 words per sentence) and 63% with summaries (5.43 sentences per summary, 4.92 words per sentence). 100% of the step and substep segments come with descriptions (4.44 words per sentence) and 15% of them include summaries (4.38 sentences per summary, 3.1 words per sentence). Figure 2 shows the distributions of auxiliary labels.

**Splits** We provide data splits for training (70%), validation (15%), and testing (15%) purposes. The splits were made at the video level, ensuring balanced inclusion of step categories across the splits. We release full annotations for the training and validation splits while withholding the test split.

## 4 Experiments

### 4.1 Tasks

**Goal/step localization** Localizing temporal action segments within a long, untrimmed video has numerous applications ranging from activity recognition to automatic chapterization for efficient navigation of instructional videos. Following temporal action localization [45], we formulate the task as predicting tuples of (`start_time`, `end_time`, `class_category`) encompassing goals and individual steps given a long untrimmed video. We use ActionFormer [56] and EgoOnly [51] as our baseline models for their superior performance demonstrated in various action localization benchmarks. We evaluate the results using the detection mAP averaged over goal *or* step categories, and over temporal IoUs {0.1, 0.2, 0.3, 0.4, 0.5}.

**Online goal/step detection** While the localization task above focuses on *offline* inference scenarios, here we consider its *online* counterpart. The task is formulated as predicting goal/step categories at each timestamp as they unfold in a streaming input video. Models must utilize information from the

| Task | Metric | Model | Val | Test |
|------|--------|-------|-----|------|
| Goal localization | Detection mAP | ActionFormer [56] | $42.2 \pm 3.8$ | $45.9 \pm 2.4$ |
| Step localization | Detection mAP | ActionFormer [56]
EgoOnly [51] | $9.9 \pm 0.3$
$13.1 \pm 0.3$ | $8.7 \pm 0.3$
$14.0 \pm 0.4$ |
| Online goal detection | Per-frame mAP | LSTR [53] | $21.5 \pm 0.5$ | $24.5 \pm 1.3$ |
| Online step detection | Per-frame mAP | LSTR [53]
EgoOnly [51] | $8.7 \pm 0.2$
$10.2 \pm 0.1$ | $7.9 \pm 0.2$
$10.8 \pm 0.1$ |
| Step grounding | Recall@1, mIoU=0.3 | VSLNet [57] | $11.7 \pm 0.2$ | $10.7 \pm 0.3$ |

Table 2: **Main results** on tasks supported by our dataset. We report standard deviation over 8 runs.

past leading up to the current timestamp, without having access to future frames. This task holds particular significance in egocentric vision due to its practical applications in AR scenarios such as real-time step-by-step guidance for procedural activities. We adopt LSTR [53] and EgoOnly [51] as our baseline models, given their strong performance and the availability of open-source implementations. Following existing literature, we report per-frame mAP.

**Step grounding**  Unlike the localization and detection tasks that assume a fixed set of categories, some scenarios require models that can recognize new, unseen steps from open vocabulary text descriptions. Thus, we study step grounding, where given a long, untrimmed egocentric video and a natural language description of a step (e.g., "beat eggs with chopsticks" in Figure 1), a model must predict its temporal extent (start_time, end_time) in the video (i.e., the green bar). We adopt VSLNet [57], a popular video query grounding model due to its strong performance on similar egocentric grounding tasks. We report Recall@1, IoU=0.3 following prior work [22]. Note that we do not consider goal grounding in this work since many videos contain a single goal spanning most of a video, for which recall measures are trivially high.

**Implementation details**  For all experiments except for the EgoOnly [51] baseline, we use pre-computed clip-level features extracted densely from each video using Omnivore [19], which are publicly available for download on the official Ego4D repository (we used "omnivore_video_swinl"). The Omnivore model has been pretrained on a combination of multiple modalities (images, videos, and 3D data) in a supervised fashion and has been demonstrated to achieve strong generalization ability across a wide variety of vision tasks. EgoOnly [51] pretrains the ViT [14] backbone from scratch on the raw frames of Ego4D [22] using the MAE [24] objective, then further finetunes it on a combination of four existing action recognition datasets (Kinetics-600 [7], Ego4D Moments [22], EPIC-Kitchens-100 [13], COIN [49]) in a supervised fashion. For EgoOnly on the online detection task, we attach a single-layer linear prediction head on top of the pretrained ViT backbone and train the entire model end-to-end. Each prediction is made on input frames with 2 second temporal context. For EgoOnly on the offline localization task, we take the ViT backbone finetuned on the online detection task and attach the ActionFormer head [56] on top, and train just the prediction head while keeping the ViT backbone frozen throughout training. For all tasks, we use open source baseline implementations[1] and adapt hyperparameters for the proposed dataset (details in Supp.).

## 4.2  Main results

Table 2 reports the main results for all the tasks on both the validation and test sets. EgoOnly [51] achieves strong performance on offline step localization and online step detection, demonstrating the effectiveness of its egocentric pretraining strategy and corroborating the strong empirical evidence presented in their paper. Note that, compared to ActionFormer [56] and LSTR [53] that leverage precomputed features from Omnivore [19], EgoOnly finetunes its ViT backbone directly on our dataset by solving the temporal segmentation task, providing further performance improvement over the two other baselines.

---

[1]Actionformer (https://github.com/happyharrycn/actionformer_release), LSTR (https://github.com/amazon-science/long-short-term-transformer), and VSLNet (https://github.com/EGO4D/episodic-memory/tree/main/NLQ/VSLNet)

| Train on | Avg. Length | Localization | | Online detect. | |
|---|---|---|---|---|---|
| | | Goal | SSteps | Goal | SSteps |
| Goals [583] | 1500 s | **42.2** | - | **21.5** | - |
| SSteps [32k] | 32 s | - | **9.9** | - | **8.7** |
| Goals + SSteps [32k] | 59 s | 37.1 | 9.7 | 21.2 | 8.1 |

\* "SSteps" denotes Steps + Substeps
\* Numbers in square bracket denotes number of training samples

| Train on | Step | Substep |
|---|---|---|
| (A) Substeps [15k] | 9.6 | 6.4 |
| (B) Steps [12k] | 13.9 | 5.8 |
| (C) Substeps + Parent steps [15k] | 7.5 | 8.1 |
| (D) Substeps [15k] + Narr. [12k] | 10.5 | 7.4 |
| (E) Steps [12k] + Narr. [15k] | 14.0 | 5.6 |
| Steps [12k] + Substeps [15k] | **15.8** | **8.7** |

Table 3: **Validation set results on hierarchical goal-step relationship**. **Left: localization and online detection.** Combining goal and step/substep instances does not help each other, likely due to the large difference in segment lengths and the limited number of goal samples. Metric: detection/per-frame mAP. **Right: grounding.** Training jointly on steps and substeps outperforms other alternatives, highlighting the benefit of the hierarchical nature of our step annotations. Metric: Recall@1, IoU=0.3. These results suggest that exploiting the step-substep hierarchy is clearly beneficial (right table), while effectively leveraging the goal-step hierarchy needs further investigation (left table).

Compared to the step prediction tasks, goal prediction requires a much longer temporal context. For instance, step segments have an average duration of 32.5 seconds, whereas goal segments have an average duration of 1946.9 seconds. This makes it a challenging long-form video understanding task. We report ActionFormer and LSTR baseline results but omit EgoOnly results due to its focus on relatively shorter-term temporal context. Specifically, EgoOnly's ViT backbone is fine-tuned with 2-second clips from our dataset, providing limited long-term context to the learned representations and, as a result, achieving inferior results compared to baselines that leverage Omnivore features. This highlights the challenging nature of our dataset and warrants further investigation into refinements in training and modeling approaches to achieve a more balanced performance across goal and step prediction tasks.

On the step grounding task, models achieve roughly similar recall scores on the validation and test splits. Note that a subset of text queries in each split belong to *step classes* that are not seen during training. For example, the step category label "Add the melted chocolate and blend until smooth" does not occur in the training set, but its corresponding natural language description "Add chocolate balls to blender jar" is still groundable. The results on this *zero-shot* subset are understandably lower – 4.3 and 3.8 Recall@1, mIoU=0.3 on validation and test splits, respectively.

### 4.3 How do models exploit the hierarchical goal-step-substep relationship?

Next, we study the hierarchical relationship among goals, steps, and substeps in our dataset. First, we explore the goal-step hierarchy by comparing models trained on goals only, steps and substeps only, and all combined. Table 3 (left) shows that simply combining all instances does not help the individual tasks of localization and online detection, likely due to the non-overlapping taxonomy, the large difference in segment lengths, as well as limited number of goal samples.

However, once we switch to step-substep hierarchy, in Table 3 (right), we find that jointly training on steps and substeps (last row) is superior to models trained on substeps only (A) or steps only (B). This joint training is significantly more effective than pairing steps with their parent descriptions (C).[2]

Importantly, the gain over (A) and (B) is not simply an effect of dataset size. In (D-E), we replace steps/substeps with an equivalent amount of Ego4D narration data from the same videos to match the total training set size of the result in the last row. The narrations capture low-level actions performed by the camera-wearer (e.g., "camera-wearer picks up a pan") and have been shown to improve performance in egocentric grounding models [39]. Our results show that while adding narrations can offer a small improvement, jointly training on steps and substeps remains a superior strategy.

Put together, the results suggest that jointly training with steps and substeps leads to models that are aware of the sequential and hierarchical relationships between steps, leading to stronger performance across both levels.

---

[2]This was done by concatenating the step and substep descriptions and using that as our hierarchy-aware query. More sophisticated, embedding-based step and substep fusion performed worse in our experiments.

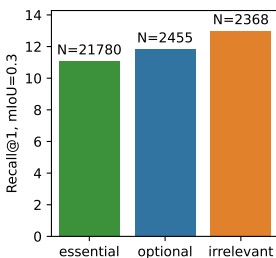

Figure 3: **Grounding performance breakdown by relevance type.** Essential steps are hardest to recognize as they tend to involve the same set of objects in related tasks (e.g., different stages of preparing the dough in various baking scenarios), requiring fine-grained recognition of objects, scenes, and actions. Optional and irrelevant steps, despite being rare during training and not as related to the activity, are visually distinct and easier to recognize. Numbers above bars denote the number of training samples in each category.

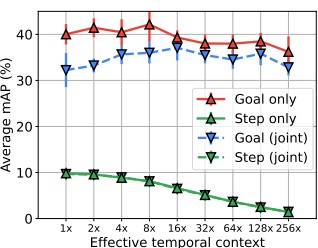
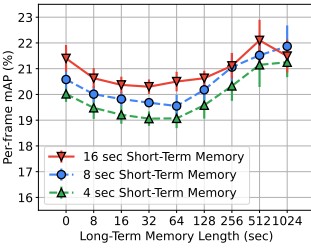
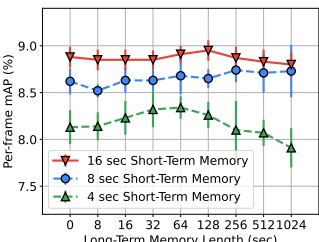

Figure 4: **Left: Goal/Step localization** with varying effective temporal context. Longer temporal context is required to disambiguate goals, while steps favor shorter context for finer temporal details. Joint training with shorter context adversely affects goal localization due to the dominance of step segments in the training signal. **Middle: Online goal detection** with varying LSTR memory length. For goals, a larger short term memory consistently leads to better results. **Right: Online step detection** with varying LSTR memory length. For steps, capturing fine-grained information via short-term temporal context is more important than capturing long-term context.

## 4.4 Which types of steps are easier to recognize?

We evaluate grounding models based on the role of steps in the overall procedure: essential, optional, or irrelevant. Essential steps are necessary for completing the task (e.g., cracking eggs for cooking an omelet), optional steps are relevant but not essential (e.g., adding sriracha sauce to an omelet), and irrelevant steps are unrelated (e.g., answering phone while cooking). Figure 3 shows that *essential* steps are the most difficult to recognize. This is somewhat counterintuitive since they occur frequently within the task (as evidenced by the largest number of instances reported in the figure) and are contextual with the activity. However, we found that they are difficult to distinguish from other steps of the related tasks as they tend to involve the same set of objects, scenes, and actions, e.g., flatten dough on table, knead dough, coat dough with flour in the task "Make bread," requiring fine-grained recognition. Conversely, optional steps (e.g., dispose eggshells, wear apron) and irrelevant steps (e.g., use phone, drink tea) are visually distinct in the context of the procedure and thus are easier to recognize, overcoming the disadvantage of relatively low training data (number above each bar) and despite their loose (or missing) connection to the goal.

## 4.5 Can long-term temporal context benefit procedural activity recognition?

Recognizing procedural activities in egocentric videos, especially in the online inference regime, requires long-term temporal context to accumulate enough past history [52]. For example, recognizing different but similar recipes may become possible only after observing completion of certain steps.

We quantitatively study this property by exploiting the hierarchical structure in Ego4D Goal-Step. In the localization task, this is achieved by adjusting the feature sampling stride before feeding features into the ActionFormer [56] that localizes goals and/or steps. This changes the effective temporal context consumed, especially when the theoretical receptive field does not cover the full video. Results are visualized in Figure 4 (left) with error bars representing standard deviation of 8 runs. We find that a larger temporal context is indeed required to disambiguate goals, while steps favor a much smaller context for finer temporal details. Additionally, we find goal-step joint training with a small context adversely affects goal localization due to the dominance of shorter steps in the training signal. As a result, the step results with and without goal instances are almost overlapping.

In online goal/step detection tasks, we follow the experimental setting of LSTR [53] that varies both short-term and long-term memory lengths. For online goal detection (Figure 4 middle), we find that increasing the long-term memory to 256 seconds and longer leads to noticeable gains. Furthermore, a larger short term memory consistently leads to better results. These findings demonstrate that predicting goals indeed requires long-term temporal context. This aligns with our intuition that disambiguating the goals requires piecing together evidence that needs to be accumulated over a long temporal span. For instance, in the case of preparing a recipe, it may only become fully recognizable after several ingredients have been used and multiple steps have been completed.

On the other hand, for online step detection (Figure 4 right), performance increases when we switch the short term memory length from 4 seconds to 8 seconds, but plateaus when we further push to 16 seconds. Also, longer long-term memory does not seem to improve performance; in fact, when a 4 second short-term memory is used, performance decreases when the long-term memory exceeds 64 seconds, which is twice as long as the average step segment duration of 32.5 seconds. This pattern is consistent with our observation on step localization, suggesting that step detection favors fine-grained information captured in short-term temporal context over longer context.

## 5 Conclusion

We presented Ego4D Goal-Step, a new set of annotations with a hierarchical taxonomy of procedural activity labels on the recently released Ego4D. It is larger than existing procedural video datasets, contains hierarchical action labels (goals - steps - substeps), and provides various auxiliary information useful for procedural activity understanding. We demonstrated three distinct task scenarios supported by our dataset – temporal localization, online detection, and grounding – and analyzed how various research questions can be explored such as hierarchical learning and long-term video modeling.

The comprehensive nature of our dataset calls for future investigations into other aspects of procedural activity understanding. One promising direction is incorporating existing Ego4D annotations for more comprehensive analyses of procedural activities. For instance, narrations and FHO/Moments provide atomic action labels that can be combined with our dataset to form a 4-level hierarchy (goals - steps - substeps - actions). Furthermore, hand & object interaction annotations provide object state information that can enable object state-based progress monitoring. We eagerly anticipate the emergence of active research threads pursuing these directions.

**Limitations and societal impact**    We acknowledge that Ego4D Goal-Step is intended for research purposes and should not be regarded as a comprehensive dataset encompassing the full range of daily human activities. Models trained on our dataset may exhibit biases towards the specific activities included in the dataset, resulting in a limited coverage of our everyday living scenarios.

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

# Appendix A   Implementation details

For all the baselines except for EgoOnly [51], we use pre-computed clip-level features extracted densely from each video using Omnivore [19], which are publicly available for download on the official Ego4D repository (we used "`omnivore_video_swinl`"). These features are extracted with a stride of 16 frames, which is 0.533s or 1.875 features per second on Ego4D videos of 30 fps.

For EgoOnly [51], we use the checkpoint which was pretrained on the raw frames of Ego4D [22] using the MAE [24] objective and then further finetuned with the temporal segmentation objective on a combination of four popular action recognition datasets: Kinetics-600 [7], Ego4D MQ [22], EPIC-Kitchens-100 [13], COIN [49]. In our experiments, we take this checkpoint and further finetune it on the temporal segmentation objective with our step annotations, for 10 epochs[3] with a learning rate of 1e-5.

**Goal/step localization.**   We use ActionFormer [56] and EgoOnly [51] as our baseline approaches. ActionFormer combines multi-scale feature representations with local self-attention, and a light-weight decoder to predict an action category at every moment in time. We use the official implementation of ActionFormer [56] for the EPIC-Kitchens-100 [13] dataset, unless noted otherwise, and tune the hyperparameters based on the best result on the validation set. We extend the FPN from 5 stages to 9 stages to learn from longer temporal context. Training takes less than 7 hours (32 epochs) on a single V100 GPU. We report the average and standard deviation of 8 runs for each result.

For EgoOnly [51], we attach ActionFormer [56] on top of the features generated from the EgoOnly backbone finetuned on temporal step segmentation. We follow the same hyperparameters in this stage as the ActionFormer baseline, i.e. 9-stage FPN, 32 epochs.

For step localization experiments ("SSteps" in Table 3), we train the models by combining step and substep instances and treating them equally. For evaluation, we average the mAP across all the step and substep categories because they share the same taxonomy. In the joint goal-step localization experiments ("Goals + Ssteps" in Table 3 and Figure 4 left), we merge the goal and step taxonomy categories and combine all of goal, step, substep instances. However, evaluation is still independent for each category, regardless of the category coming from the goal taxonomy or the step taxonomy, following the common detection per-class mAP measure. Then, the goal localization results are averaged as the goal mAP, and the step localization results are averaged as step mAP.

We use the base learning rate of 2e-4 and train the model for 32 epochs, with linear warm-up for 16 epochs. We denote the raw feature stride of 0.533s (i.e. 1.875 fps) by the $1\times$ "effective temporal context" (ETC) reported in Figure 4 left of the main paper. Based on the best results reported in Figure 4 (left), we set the ETC for step localization to be $1\times$, for both step-only and goal-step joint experiments. For goal localization, we set it to $8\times$ (feature stride of 4.266s) for goal-only experiments and $16\times$ (feature stride of 8.533s) for goal-step joint experiments.

**Online goal/step detection.**   We use LSTR [53] and EgoOnly [51] as our baseline approaches. LSTR combines an encoder that captures the long-term coarse-scale information and a decoder that models short-term information. We use the official implementation of LSTR [53], including the default short-term memory of 8 seconds and the long-term memory of 512 seconds, unless noted otherwise. In our experiments, we found that training the model for 4 epochs yields the best performance on the validation split. We train with the base learning rates of 4e-5 for goals, and 2e-4 for steps. We follow the default hyperparameters provided in the official implementation, and apply the linear learning rate warm-up for the first 40% of the training iterations. Training each model takes less than 4 hours (4 epochs) on a single V100 GPU.

For EgoOnly, we attach a single-layer classification head on top of the pretrained EgoOnly backbone and train the entire network end-to-end with the temporal segmentation objective with the default temporal context of 2 seconds. For online step detection inference and evaluation, we take only the last frame classification output as the prediction at the current timestamp. We use the same model to extract features for offline step localization experiments.

**Step grounding.**   For this task, we use all annotated instances of steps and substeps. We ignore goal segment annotations as they do not inform model selection. This is because many videos contain

---

[3]We note that here the notion of epoch is different from ActionFormer/LSTR because of the differences in training settings; we refer to EgoOnly [51] for details.

a single goal that spans a large portion of the video, which results in trivially high recall scores (as mentioned in Section 4.1).

For our baseline model, we use VSLNet [57]. The model works as follows. First, a sequence of features are extracted for the video and text query. A cross-attention mechanism then aggregates information across modalities to produce one feature per time-step. Finally, two LSTM models aggregate these features to predict the start / end time probabilities at each time-step. We refer the reader to the original paper for full model details [57].

Each video is re-sampled to 128 features for training. We train models for 200 epochs with learning rate 1e-3, batch size 32, and select the model with the highest validation set performance for testing. The remaining hyperparamters follow the default for Ego4D NLQ [22]. Training takes approximately 6 hours on two V100 GPUs. We report an average of 8 runs for each result (Table 2 in the main paper).

## Appendix B    Additional dataset statistics

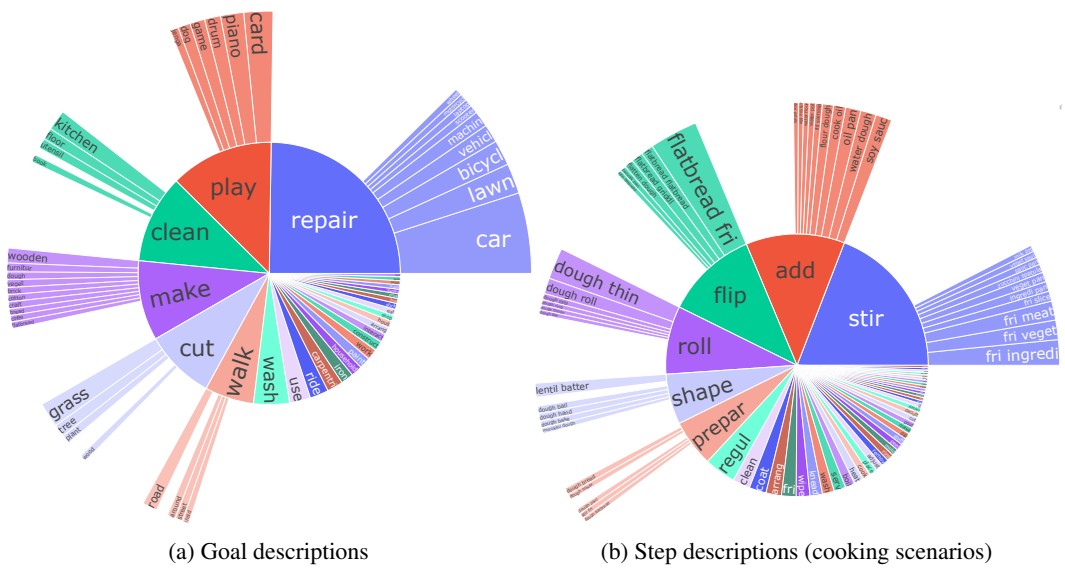

(a) Goal descriptions        (b) Step descriptions (cooking scenarios)

Figure 5: **Distribution of natural language descriptions for goals and steps.**

Figure 5 illustrates the distribution of natural language descriptions for goals and steps/sub-steps combined. To create the plot, we removed stop words and stemmed the remaining words. For the goal descriptions, we display the first two words, while for the step/sub-step descriptions, we display the first three words.

## Appendix C    Datasheet for Ego4D Goal-Step

We provide the datasheet following the format suggested by Gebru et al. [16]. We would like to note that Ego4D Goal-Step contains new annotations on *existing* videos in Ego4D. No new videos were collected or recorded during the annotation process. In all our responses below, the term "data" specifically refers to the *annotations*, not the videos associated with Ego4D Goal-Step, unless otherwise noted.

### C.1    Motivation

a) **For what purpose was the dataset created?** Was there a specific task in mind? Was there a specific gap that needed to be filled? Please provide a description.

Ego4D Goal-Step was created to support research on human activity understanding in long-form egocentric videos. Specifically, it provides procedural activity labels organized in a hierarchical

structure, capturing three distinct granularities of human actions (goals - steps - substeps). It supports various tasks in procedural activity understanding, including temporal action localization [56] and temporal grounding of step descriptions [58] – both of which we demonstrate in our paper – as well as other related tasks including hierarchical action segmentation [17], action anticipation [18], procedure planning [9], and task graph learning [31].

b) **Who created the dataset (e.g., which team, research group) and on behalf of which entity (e.g., company, institution, organization?**

Ego4D Goal-Step was created by Fundamental AI Research (FAIR) at Meta. The authors of this paper are also part of the Ego4D consortium.

c) **Who funded the creation of the dataset?** If there is an associated grant, please provide the name of the grantor and the grant name and number.

This project is funded by Meta.

## C.2 Composition

a) **What do the instances that comprise the dataset represent (e.g., documents, photos, people, countries)?** Are there multiple types of instances (e.g., movies, users, and ratings; people and interactions between them; nodes and edges)? Please provide a description.

Each instance is a set of activity annotations on a particular video in Ego4D [22], organized in a hierarchical form. We show an example annotation in Listing 1.

b) **How many instances are there in total (of each type, if appropriate)?**

Ego4D Goal-Step is comprised of two main components.

- **The goal annotation set** consists of 7,353 videos, totaling 2,807 hours. This set covers 72% of Ego4D after discarding non-procedural or uninteresting videos, and is intended to provide a large-scale training and evaluation dataset for goal inference.
- **The step annotation set** focuses on the cooking scenario portion of Ego4D, and is intended specifically for procedural activity recognition. It consists of 47,721 densely labeled segments, amounting to 430 hours in total.

Listing 1: Example annotation in a JSON format

```
{
"9b58e3ab-7b6d-4e79-9eea-c21420b0eedc": {
    "start_time": 0.0210286458333333,
    "end_time": 510.1876953125,
    "goal_category": "COOKING:MAKE_OMELET",
    "goal_description": "Make omelette",
    "goal_wikihow_url": "https://www.wikihow.com/Cook-a-Basic-Omelette
        ",
    "summary": [
        "Toasting bread on a pan",
        "Making omelet",
        "Serving omelet with ketchup"
    ],
    "is_procedural": true,
    "segments": [
    {
        "start_time": 0,
        "end_time": 56.99209,
        "step_category": "General cooking activity: Toast bread"
        "step_description": "Toast bread",
        "is_continued": false,
        "is_procedural": true,
        "is_relevant": "essential",
        "summary": [
            "heat skillet",
            "toast bread",
            "trash kitchen waste"
```

```
        ],
        "segments": [
        {
            "start_time": 0,
            "end_time": 13.135,
            "step_category": "Cook on a stovetop: Turn on the stovetop
                "
            "step_description": "preheat the stove-top",
            "is_continued": false,
            "is_relevant": "essential",
            "is_procedural": true,
            "summary": [
                "turn on stove",
                "preheat the stove-top"
            ],
            "segments": [],
        },
        ...
        ]
        ...
    }
    ]
}
```

---

c) **Does the dataset contain all possible instances or is it a sample (not necessarily random) of instances from a larger set?** If the dataset is a sample, then what is the larger set? Is the sample representative of the larger set (e.g., geographic coverage)? If so, please describe how this representativeness was validated/verified. If it is not representative of the larger set, please describe why not (e.g., to cover a more diverse range of instances, because instances were withheld or unavailable).

The goal annotation set covers 72% of Ego4D. We filtered out videos based on scenarios that are deemed non-procedural or uninteresting. The discarded scenarios are:

- **Eating**: ''Eating'', ''Eating at a restaurant'', ''Eating in a canteen'', ''Eating in hawker center''
- **Talking**: ''Talking with family members'', ''Talking with friends/housemates'', ''Talking to colleagues'', ''Talking on the phone''
- **Attending a meeting**: ''Attending a TA session'', ''Participating in a meeting''
- **Watching something**: ''Watching tv'', ''Working at desk'', ''Reading books'', ''Video call'', ''Playing games/video games'', ''On a screen (phone/laptop)'', ''Play with cellphone''
- **Commuting / moving around**: ''Bike'', ''Cycling/jogging'', ''Car-commuting, road trip'', ''Skateboard/scooter'', ''Walking the dog/pet'', ''Walking on street'', ''Clothes, other shopping''

The step annotation set contains only cooking scenario videos due to their strong procedural characteristics.

Ego4D metadata includes information about collection sites (university names), which contains geographical information. This allows us to analyze the geographical diversity of the goal and step annotation sets. The distribution of collection sites are (see Figure 6):

- **Ego4D**: 'cmu': 1898, 'unict': 1457, 'iiith': 1233, 'minnesota': 1123, 'kaust': 960, 'frl_track_1_public': 836, 'utokyo': 780, 'bristol': 778, 'cmu_africa': 144, 'uniandes': 137, 'nus': 115, 'indiana': 95, 'georgiatech': 55.
- **Goal annotation set**: 'cmu': 1675, 'unict': 1214, 'utokyo': 840, 'iiith': 832, 'minnesota': 701, 'frl_track_1_public': 606, 'bristol': 529, 'kaust': 524, 'cmu_africa': 144, 'nus': 115, 'uniandes': 74, 'indiana': 62, 'georgiatech': 37.
- **Step annotation set**: 'utokyo': 237, 'iiith': 204, 'bristol': 101, 'minnesota': 96, 'unict': 77, 'kaust': 77, 'cmu_africa': 41, 'nus': 7.

We can see the the goal annotation set closely follows the distribution of the entire Ego4D dataset. The step annotation set is different from those distributions. We note that this is because cooking

scenario videos were collected only in those collection sites, rather than us intentionally selecting videos from certain collection sites.

**d) What data does each instance consist of? "Raw" data (e.g., unprocessed text or images) or features? In either case, please provide a description.**

Each instance consists of time interval, goal/step category and description, text summary, and various other auxiliary information. For full details of the information contained in each instance, please see Listing 1.

**e) Is there a label or target associated with each instance?** If so, please provide a description. There are various labels associated with each instance. Below shows the definition of each field.

- `start_time` and `end_time`: Time interval
- `goal_category` and `step_category`: goal and step category label
- `goal_description` and `step_description`: goal and step description in natural language
- `goal_wikihow_url`: A wikiHow URL that best describes the activity captured in a video
- `summary`: A bullet-pointed summary of steps contained in each segment
- `segments`: A list of step segments (or substep segments) under a given goal (or step) segment
- `is_procedural`: A boolean flag indicating whether this segment contains procedural activity.
- `is_continued`: A boolean flag indicating whether this segment is a continuation of an activity (goal, step, or substep) from the most recent earlier segment of the same activity.
- `is_relevant`: One of "essential", "optional", "irrelevant" indicating how relevant this segment is to its parent segment.

**f) Is any information missing from individual instances?** If so, please provide a description, explaining why this information is missing (e.g., because it was unavailable). This does not include intentionally removed information, but might include, e.g., redacted text.

Everything that the annotators have provided is included. No data is missing.

**g) Are relationships between individual instances made explicit (e.g., users' movie ratings, social network links)?** If so, please describe how these relationships are made explicit.

Each individual video in Ego4D contains metadata about collection site (university name), from which relationships can be established across individual instances. This information is available in the Ego4D dataset. In Ego4D Goal-Step, to avoid redundancy, we do not explicitly provide the collection site information, but this can be easily retrieved by using the unique video identifier (the keys in the JSON dictionary, e.g., `"9b58e3ab-7b6d-4e79-9eea-c21420b0eedc"` in Listing 1).

**h) Are there recommended data splits (e.g., training, development/validation, testing)?** If so, please provide a description of these splits, explaining the rationale behind them.

We provide data splits for training, validation, and testing purposes. We release full annotations for the training and validation splits while withholding the test split. To facilitate evaluation, we will set up and maintain a test server on EvalAI. Participants can upload their results to the server, where they will be evaluated automatically.

We divide the data into training (70%), validation (15%), and test (15%) splits. The split is performed at the video level to ensure no information leakage across splits. To achieve this, we employed stratified sampling over videos using the `train_test_split` function from the `sklearn` Python package. Each video is assigned a single category label. We adopted a greedy assignment approach, iterating over step categories sorted by their frequency. Step labels are assigned to videos until a minimum number is reached, which we set at a minimum of 4 instances per step category.

**i) Are there any errors, sources of noise, or redundancies in the dataset?** If so, please provide a description.

As described in Section 3.2 of the main paper, annotators provide both category labels and natural language descriptions for segments of goals, steps, and substeps identified in a video. When annotators cannot find a category label in the taxonomy, they select `"other"` and suggest a new category in the description. We periodically review those descriptions labeled with `"other"` and either map them to existing categories or add them to the taxonomy. This is done through keyword mapping and manual verification. Due to the manual mapping process and the inherent complexity of human activity, it is possible that the process is noisy.

j) **Is the dataset self-contained, or does it link to or otherwise rely on external resources (e.g., websites, tweets, other datasets)?** If it links to or relies on external resources, a) are there guarantees that they will exist, and remain constant, over time; b) are there official archival versions of the complete dataset (i.e., including the external resources as they existed at the time the dataset was created); c) are there any restrictions (e.g., licenses, fees) associated with any of the external resources that might apply to a dataset consumer? Please provide descriptions of all external resources and any restrictions associated with them, as well as links or other access points, as appropriate.

Ego4D Goal-Step is a new set of annotations on existing videos in Ego4D, and will be hosted directly on the official Ego4D repository alongside other annotations.

The repository itself is supported by Meta and the 15 university Ego4D consortium, and the board and consortium exist independent of Meta. There is funded work already underway for future versions of the dataset and no doubts about the long-term persistence of the consortium or the flagship video dataset itself which these annotations are based upon.

Each version of the dataset that is updated is marked as an individual update, and prior versions are always available in their original form via the CLI. The dataset will also include appropriate metadata to confirm the correct versions and videos are downloaded and used beyond that guarantee.

Ego4D Goal-Step annotations are available via the CLI and do not require a license or any restrictions (once publicly released). For the Ego4D videos themselves, there is no cost, but an individual or entity must sign the license and receive approval to download the access keys as detailed above. Requests are only rejected for incorrect submissions or from being from a country with comprehensive US, UK or EU trade restrictions (currently Crimea, Donetsk, and Luhansk regions of Ukraine, Russia, Cuba, North Korea, Iran, and Syria - though that is subject to change).

k) **Does the dataset contain data that might be considered confidential (e.g., data that is protected by legal privilege or by doctor–patient confidentiality, data that includes the content of individuals' non- public communications)?** If so, please provide a description.

No information contained in Ego4D Goal-Step is considered confidential.

l) **Does the dataset contain data that, if viewed directly, might be offensive, insulting, threatening, or might otherwise cause anxiety?** If so, please describe why.

No.

m) **Does the dataset identify any subpopulations (e.g., by age, gender)?** If so, please describe how these subpopulations are identified and provide a description of their respective distributions within the dataset.

No explicit process was in place to identify any subpopulations during the annotation process. The activity labels were indiscriminate of the identity of the camera wearer who recorded the videos.

n) **Is it possible to identify individuals (i.e., one or more natural persons), either directly or indirectly (i.e., in combination with other data) from the dataset?** If so, please describe how.

It is not possible to identify individuals from the labels provided in Ego4D Goal-Step. All videos in Ego4D went through a set of de-identification processes to ensure a high standard of data privacy, see Appendix B of Grauman et al. [22] for details.

o) **Does the dataset contain data that might be considered sensitive in any way (e.g., data that reveals race or ethnic origins, sexual orientations, religious beliefs, political opinions or union memberships, or locations; financial or health data; biometric or genetic data; forms of government identification, such as social security numbers; criminal history)?** If so, please provide a description.

No.

## C.3 Collection Process

a) **How was the data associated with each instance acquired?** Was the data directly observable (e.g., raw text, movie ratings), reported by subjects (e.g., survey responses), or indirectly inferred/derived from other data (e.g., part-of-speech tags, model-based guesses for age or language)? If the data was

reported by subjects or indirectly inferred/derived from other data, was the data validated/verified? If so, please describe how.

The videos were directly observable by the annotators, through a specialized tool with a video player and input fields tailored for our annotation (see the annotation user interface in Figure 7).

**b) What mechanisms or procedures were used to collect the data (e.g., hardware apparatuses or sensors, manual human curation, software programs, software APIs)?** How were these mechanisms or procedures validated?

We used an internal tool developed at Meta (Figure 7) for web-based annotation. The same tool was used for the creation of the Ego4D dataset, validating its robustness and scalability.

**c) If the dataset is a sample from a larger set, what was the sampling strategy (e.g., deterministic, probabilistic with specific sampling probabilities)?**

See our response in D.2.c.

**d) Who was involved in the data collection process (e.g., students, crowdworkers, contractors) and how were they compensated (e.g., how much were crowdworkers paid)?**

We contracted a third party vendor to manage the annotation process, and checked privacy and ethical compliance through rigorous reviews. The amount of compensation were determined via contract, and the payments were made on a monthly basis. Because of multiple annotation projects involved with the vendor, the exact breakdown for how much they were paid is unknown to us.

**e) Over what timeframe was the data collected? Does this timeframe match the creation timeframe of the data associated with the instances (e.g., recent crawl of old news articles)?** If not, please describe the timeframe in which the data associated with the instances was created.

The entire annotation took 6 months to collect, from late 2022 to mid 2023. The annotation timeframe does not match the creation timeframe of video recordings, which was made in year 2021 through 2022.

**f) Were any ethical review processes conducted (e.g., by an institutional review board)?** If so, please provide a description of these review processes, including the outcomes, as well as a link or other access point to any supporting documentation.

Yes. This annotation project went through a rigorous internal review process at Meta for privacy and ethical compliance.

**g) Did you collect the data from the individuals in question directly, or obtain it via third parties or other sources (e.g., websites)?**

Collection or recording of new videos was not as part of Ego4D Goal-Step annotation. The annotations were provided through a third party vendor who managed contracted workers. The vendor managed all contractors, applying their well-established processes for annotator recruitment, training, auditing, and management. The contractors use our web-based annotation tool (Figure 7) to annotate videos.

**h) Were the individuals in question notified about the data collection?** If so, please describe (or show with screenshots or other information) how notice was provided, and provide a link or other access point to, or otherwise reproduce, the exact language of the notification itself.

Yes, our annotators are contracted workers who fully comprehend the nature of their tasks. They have been duly informed that we are gathering annotations from them, and these annotations will be made available to the public.

**i) Did the individuals in question consent to the collection and use of their data?** If so, please describe (or show with screenshots or other information) how consent was requested and provided, and provide a link or other access point to, or otherwise reproduce, the exact language to which the individuals consented.

Yes, see above.

**j) If consent was obtained, were the consenting individuals provided with a mechanism to revoke their consent in the future or for certain uses?** If so, please provide a description, as well as a link or other access point to the mechanism (if appropriate).

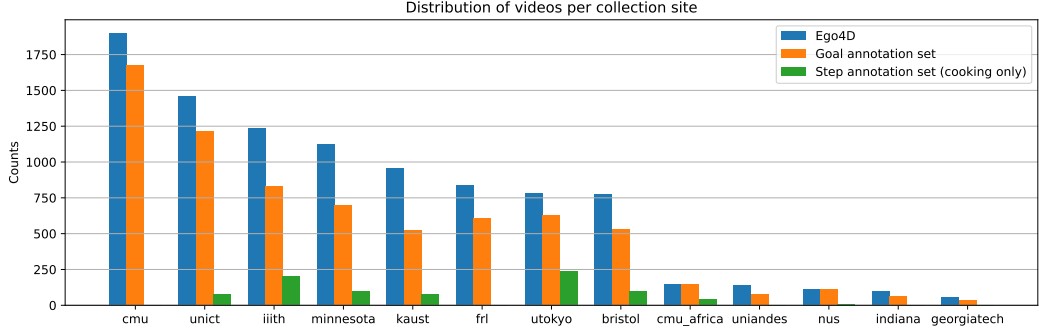

Figure 6: Comparison of geographical distributions of videos on three different sets: the original Ego4D [22], our goal annotation set, and our step annotation set. The goal annotation set follows the original distribution, while the step annotation set differs because of its focuses cooking scenarios.

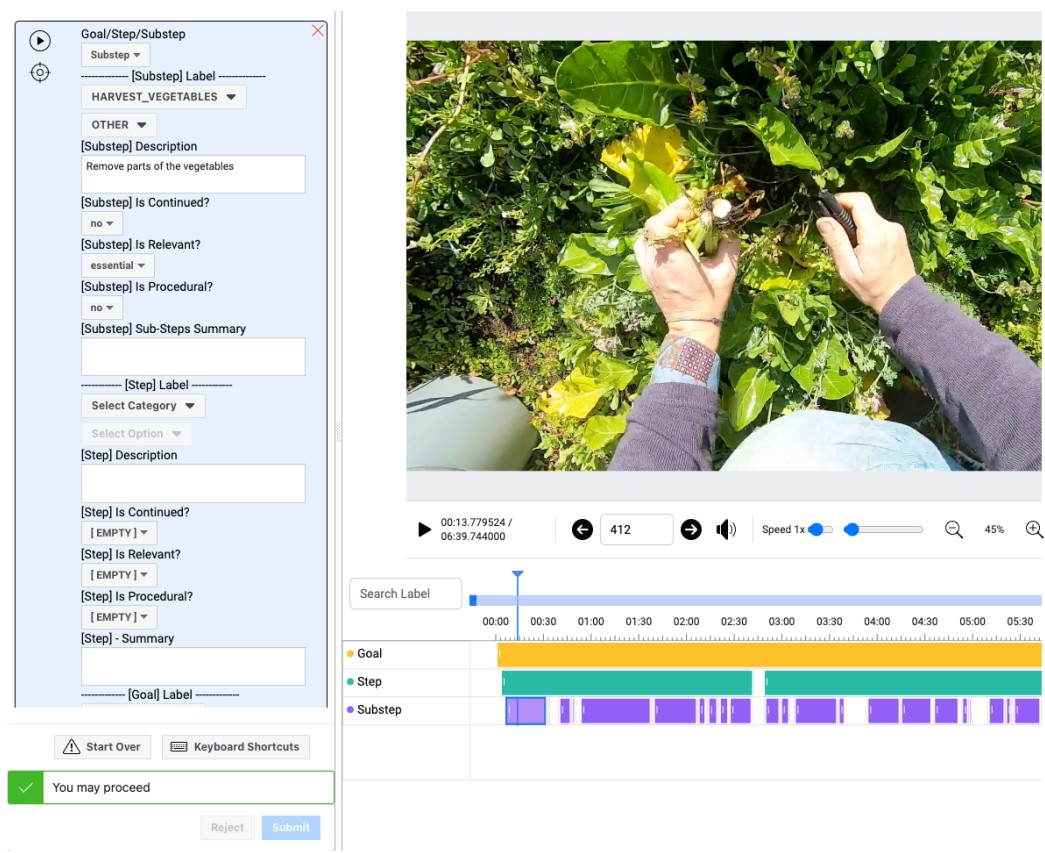

Figure 7: **The user interface used for data annotation.**

No.

**k) Has an analysis of the potential impact of the dataset and its use on data subjects (e.g., a data protection impact analysis) been conducted?** If so, please provide a description of this analysis, including the outcomes, as well as a link or other access point to any supporting documentation.

No.

### C.4 Processing, cleaning, labeling

**a) Was any preprocessing/cleaning/labeling of the data done (e.g., discretization or bucketing, tokenization, part-of-speech tagging, SIFT feature extraction, removal of instances, processing of missing values)?** If so, please provide a description. If not, you may skip the remaining questions in this section.

We mapped natural language descriptions of goals, steps, and substeps to their corresponding categories via keyword mapping followed by manual verification; see D.2.i and Section 3.2 of the main paper for details.

**b) Was the "raw" data saved in addition to the preprocessed/cleaned/labeled data (e.g., to support unanticipated future uses)?** If so, please provide a link or other access point to the "raw" data.

Yes. We also release the "raw" natural language descriptions in in Ego4D Goal-Step; see ``goal_description'' and ``step_description in Listing 1

**c) Is the software that was used to preprocess/clean/label the data available?** If so, please provide a link or other access point.

N/A.

### C.5 Uses

**a) Has the dataset been used for any tasks already?** If so, please provide a description.

Ego4D Goal-Step is not public yet, so no other papers have used it for their tasks. However, our main paper demonstrates how it can be used in three scenarios: temporal goal/step localization, online goal/step detection, and step grounding. See Section 4 for details.

**b) Is there a repository that links to any or all papers or systems that use the dataset?** If so, please provide a link or other access point.

N/A.

**c) What (other) tasks could the dataset be used for?**

Besides the tasks we demonstrate in the paper, Ego4D Goal-Step can support other tasks in procedural video understanding, such as hierarchical action segmentation [17], action anticipation [18], procedure planning [9], and task graph learning [31], and video summarization [38].

**d) Is there anything about the composition of the dataset or the way it was collected and preprocessed/cleaned/labeled that might impact future uses?** For example, is there anything that a dataset consumer might need to know to avoid uses that could result in unfair treatment of individuals or groups (e.g., stereotyping, quality of service issues) or other risks or harms (e.g., legal risks, financial harms)? If so, please provide a description. Is there anything a dataset consumer could do to mitigate these risks or harms?

No.

**e) Are there tasks for which the dataset should not be used?** If so, please provide a description.

We acknowledge that Ego4D Goal-Step is intended for research purposes and should not be regarded as a comprehensive dataset encompassing the full range of daily human activities. Similar to other large-scale real-world datasets, Ego4D Goal-Step exhibits skewed distribution of activities, subjects, and settings, despite the best effort in Ego4D to capture real-world authenticity and comprehensive diversity. As such, models trained on our dataset may exhibit biases towards the specific activities included in the dataset, resulting in a limited coverage of our everyday living scenarios.

## C.6    Distribution

**a) Will the dataset be distributed to third parties outside of the entity (e.g., company, institution, organization) on behalf of which the dataset was created?** If so, please provide a description.

Yes. The dataset will be publicly available on the internet.

**b) How will the dataset will be distributed (e.g., tarball on website, API, GitHub)?** Does the dataset have a digital object identifier (DOI)?

Ego4D Goal-Step will be distributed through the official Ego4D download protocol, which can be found in `https://ego4d-data.org`. There will be no DOI assigned specifically for Ego4D Goal-Step.

**c) When will the dataset be distributed?**

We will release Ego4D Goal-Step around before December 2023.

**d) Will the dataset be distributed under a copyright or other intellectual property (IP) license, and/or under applicable terms of use (ToU)?** If so, please describe this license and/or ToU, and provide a link or other access point to, or otherwise reproduce, any relevant licensing terms or ToU, as well as any fees associated with these restrictions.

Ego4D Goal-Step will be distributed under the standard Ego4D license agreement, which can be found in `https://ego4ddataset.com/`.

**e) Have any third parties imposed IP-based or other restrictions on the data associated with the instances?** If so, please describe these restrictions, and provide a link or other access point to, or otherwise reproduce, any relevant licensing terms, as well as any fees associated with these restrictions.

No.

**f) Do any export controls or other regulatory restrictions apply to the dataset or to individual instances?** If so, please describe these restrictions, and provide a link or other access point to, or otherwise reproduce, any supporting documentation.

No.

## C.7    Maintenance

**a) Who will be supporting/hosting/maintaining the dataset?**

The Ego4D consortium will support/host/maintain it.

**b) How can the owner/curator/manager of the dataset be contacted (e.g., email address)?**

The primary contact is Yale Song (`yalesong@meta.com`).

**c) Is there an erratum?** If so, please provide a link or other access point.

Not so far, it has not been released yet.

**d) Will the dataset be updated (e.g., to correct labeling errors, add new instances, delete instances)?** If so, please describe how often, by whom, and how updates will be communicated to dataset consumers (e.g., mailing list, GitHub)?

Yes, we plan to update the dataset with labeling corrections and newly annotated labels in the future. The updates will be made on the official Ego4D website, `https://ego4d-data.org/docs/updates/`.

**e) If the dataset relates to people, are there applicable limits on the retention of the data associated with the instances (e.g., were the individuals in question told that their data would be retained for a fixed period of time and then deleted)?** If so, please describe these limits and explain how they will be enforced.

No.

**f) Will older versions of the dataset continue to be supported/hosted/maintained?** If so, please describe how. If not, please describe how its obsolescence will be communicated to dataset consumers.

Yes, older versions will continue to be supported/hosted/maintained through the Ego4D repository.

**g) If others want to extend/augment/build on/contribute to the dataset, is there a mechanism for them to do so?** If so, please provide a description. Will these contributions be validated/verified? If so, please describe how. If not, why not? Is there a process for communicating/distributing these contributions to dataset consumers? If so, please provide a description.

Yes, we welcome community contributions through Github and any other forms of communication. These contributions will be validated and verified by us before merged into the dataset. We acknowledge community contributors in our updates, e.g., see an example of our acknowledgement in `https://ego4d-data.org/docs/updates/#nlq-annotation-updates`.

