# OpenReview forum: "Ego4D Goal-Step: Toward Hierarchical Understanding of Procedural Activities"
_NeurIPS.cc/2023/Track/Datasets_and_Benchmarks — NeurIPS 2023 Datasets and Benchmarks Spotlight_

### Official Review · Reviewer_hMCZ · 2023-07-17
**Reviews for 574**

**Rating:** 7
**Confidence:** 3
**Correctness:** Yes
**Clarity:** Yes.

**Strengths:**

1. The label scale is large.
2. For annotations, author design careful procedure to assure the label quality.
3. The motivation of this paper is interesting.
4. Human activity goal is worthy to be focused.

**Additional Feedback:**

See the above.

**Documentation:**

Yes

**Limitations:**

Yes.

**Opportunities For Improvement:**

1.The tested methods are limited. Only few works are evaluated.
2. How to unify the goal and activity in a framework?
3. More chanllenges for this tasks should be discussed.

**Relation To Prior Work:**

Yes

**Summary And Contributions:**

This paper provides hierarchical labels for Ego4d including goals, steps, substeps, which is for understanding human acitivities via object goal. The scale of dataset is large and the aims of this dataset is different from previous works.

---

> ### Author Response · Authors · 2023-08-21
> **Thank you!**
>
> Thank you for the positive feedback!
>
> Comment) ```The tested methods are limited. Only a few works are evaluated.```
>
> Thank you for the suggestion. We plan to incorporate an additional baseline method, EgoOnly [Wang et al., 2023] (https://arxiv.org/abs/2301.01380), for both the localization and online detection tasks. The model is currently achieving SoTA results on various egocentric activity detection benchmark, which is why we believe it would be a strong and interesting baseline to include.
>
> Comment) ```How to unify the goal and activity in a framework?```
>
> This is an excellent research topic on its own. There is not yet an established solution emerging from the literature. We believe that our dataset will provide unique opportunities to explore various ideas along this direction, e.g., goal-conditioned inference, step localization/detection with/without goal knowledge and/or inference. We look forward to seeing exciting developments enabled by our dataset on this front!
>
> Comment) ```More challenges for these tasks should be discussed.```
>
> We have added a paragraph in Section 5 outlining more challenges worth visiting in the future (the second paragraph, highlighted in blue).

---

### Official Review · Reviewer_1pxF · 2023-07-22
**Initial review**

**Rating:** 7
**Confidence:** 3
**Correctness:** Yes.
**Clarity:** Yes.

**Strengths:**

1) Hierarchical action labels are provided for large-scale egocentric procedure videos to support more comprehensive human activity understanding.
2) Extensive experiments are conducted to demonstrate the usefulness of the new annotations for temporal action detection and text grounding tasks.

**Additional Feedback:**

NA.

**Documentation:**

Documentation on the organization, availability of the dataset can be improved. Currently only action labels are provided.

**Ethics:**

NA.

**Limitations:**

Yes.

**Opportunities For Improvement:**

Overall, I don't find major drawback from this paper. One possible way for improvement might be to provide more information on how to access and use the data of Ego4D Goal-Step. Currently only annotation files are provided.

**Relation To Prior Work:**

Yes.

**Summary And Contributions:**

This paper proposes a video dataset (Ego4D Goal-Step) for hierarchical understanding of human activities. The data is based on procedure videos of the public Ego4D dataset. The main contribution is the hierarchical action labels (goal, step, sub-step) for these videos. In addition, comprehensive experiments and analysis are done to demonstrate how Ego4D Goal-Step supports exploring various tasks in procedural activity understanding.

---

> ### Author Response · Authors · 2023-08-21
> **Thank you!**
>
> We appreciate your positive feedback!
>
> Comment) ```provide more information on how to access and use the data of Ego4D Goal-Step```
>
> We will set up a project website providing detailed information on how to access the dataset, and update the paper with the URL for easy access. In short, the dataset will be provided via Ego4D CLI (https://ego4d-data.org/docs/CLI/) which allows for easy access to videos, precomputed features, and annotations for all the benchmarks currently available.

---

### Official Review · Reviewer_iAEY · 2023-07-23
**A hierarchically procedural annotation for human activity on Ego4D**

**Rating:** 5
**Confidence:** 3

**Strengths:**

- This work provides goal-oriented annotation for human activities. This is very important to understand the high-level goal or objective the sequences of actions are aiming to achieve.
- Ego4D Goal-Step is the largest available egocentric dataset of annotations for procedural activity understanding.
- In addition to goal-step-substep annotation, it also provides auxiliary annotation, such as summary and goal-oriented labels.
- The experiment results show that jointly training with steps and substeps leads to models that are aware of the sequential and hierarchical relationships between steps, leading to stronger performance across both levels.

**Additional Feedback:**

No additional feedback.

**Clarity:**

This paper is well-written and easy to read.
- L55: duplicate “to”

**Correctness:**

- L200: “This suggests that most of the step/substep segments capture higher-level actions than [22, 34] without sacrificing annotation density or capturing short-term atomic actions.” There is no clue for this conclusion in the context. More explanation is needed for getting this claim.
- After reading the main paper and supplemental, the actual data annotation process is still not clearly stated. For example, how to select or train the annotators, how many iterations were needed, and when to do the many taxonomy updates.

**Documentation:**

After reading the main paper and supplemental, the actual data annotation process is still not clearly stated. For example, how to select or train the annotators, how many iterations were needed, and when to do the many taxonomy updates.
Other than this, the dataset has the details for all requirements.

**Ethics:**

The authors did not provide hourly wages and considered it confidential.

**Limitations:**

Yes, the authors adequately addressed the limitations and potential negative societal impact of their work. The authors mentioned the limitations of biases towards certain activities such as cooking videos instead of being a comprehensive dataset capturing the full range of human daily activities.

**Opportunities For Improvement:**

- L35: in this paper, it talked about the importance of hierarchy from top to bottom: goals, steps, substeps, and atomic actions. But this paper focuses on the idea of the top level, the goal annotations, and did not add the atomic actions. This makes the proposed annotations not cover the full hierarchy.
- How did the authors keep track of the quality of the annotation from 3rd-part workers?
- L158: The goal taxonomy is done iteratively and manually. Who is defining the goals, the authors or the workers? How did the authors choose to stop iterating? How did the authors verify and access the correctness of the process?
- L162: the taxonomy initialization is relying on the LLaMA model. How to make sure the model is not biased and is correct? How to prompt the model to give the right granularity level of steps, which is too detailed or too broad?
- L200: “This suggests that most of the step/substep segments capture higher-level actions than [22, 34] without sacrificing annotation density or capturing short-term atomic actions.” There is no clue for this conclusion in the context. More explanation is needed for getting this claim.
- Fig 4 middle: the figure is very interesting, in that performance decreases as the long-term memory length is close to 128. More analysis and discussion about this phenomenon are needed for this.
- After reading the main paper and supplemental, the actual data annotation process is still not clearly stated. For example, how to select or train the annotators, how many iterations were needed, and when to do the many taxonomy updates.

**Relation To Prior Work:**

Yes, the paper discusses its difference from prior works both in the intro and related works. This paper focuses on the large-scale hierarchy of actions while previous works are relatively small-scale and focus on steps or atomic actions. The taxonomy is data-driven compared with other datasets that rely on external resources to develop taxonomies.

**Summary And Contributions:**

This paper introduces a large hierarchically annotated procedural human activities in the existing Ego4D dataset. Human activities contain primary goals on the top level, then the sequences of steps, and finally the atomic actions on the bottom. While current activity research mainly focuses on the steps and atomic actions in trimmed very short clips, this work introduces a novel hierarchy of taxonomy of goal-oriented activity labels, goal-step-substep. The size of the proposed dataset is substantially large. It provides dense annotations for 47K procedural step segments (442 hours) and high-level goal annotations for 2742 hours of Ego4D videos. It provides auxiliary information along with the annotation, which are language summary, step completion status, and step-to-goal relevance information. The taxonomy is data-driven instead of pre-defined. This dataset is useful in exploring procedural activity understanding and it demonstrated on goal/step localization, online goal/step detection, and step grounding tasks.

---

> ### Author Response · Authors · 2023-08-21
> **Appreciate the detailed and constructive feedback! (response 1/2)**
>
> We appreciate your detailed and constructive feedback.
>
> Comment) ```L35: in this paper, it talked about the importance of hierarchy from top to bottom: goals, steps, substeps, and atomic actions. But this paper focuses on the idea of the top level, the goal annotations, and did not add the atomic actions. This makes the proposed annotations not cover the full hierarchy.```
>
> The reviewer is correct that our annotation does not cover atomic actions. This was intentional, because, as we described these in Section 3.1, Ego4D already provides them via natural language descriptions (narrations) and categorical labels (FHO, moments). To make this point clear, we have added a paragraph in Section 5 (highlighted in blue) highlighting the use of narrations and FHO/Moments as one of possible ways to further explore our dataset.
>
> To make narrations & FHO/Moments easily accessible in our goal-step annotation, we can offer an option to download goal-step with narrations & FHO/Moments through our Ego4D CLI.
>
> Comment) ```How did the authors keep track of the quality of the annotation from 3rd-party workers?```
>
> Our 3rd-party vendor has an auditing team reviewing the quality of individual annotations. 100% of the annotations were reviewed and about 10% were re-annotated as part of the auditing process. Throughout the annotation process, we held weekly meetings with their management team to discuss labeling issues and shared with them further documentation to resolve them.
>
> Comment) ``` L158: The goal taxonomy is done iteratively and manually. Who is defining the goals, the authors or the workers? How did the authors choose to stop iterating? How did the authors verify and access the correctness of the process?```
>
> It was the authors who defined the goal categories. As described in Section 3.2 – Stage 2, for missing goal categories, annotators chose “other” and described it in free-form text. The authors then reviewed the annotations in batches, mapping keywords in free-form text to existing goal categories and also adding new categories if they were not already in the taxonomy. We also manually verified the correctness of keyword mapping by visual inspection. This involved visualizing the goal categories, free-form descriptions, and thumbnails of the video.
>
> The data was annotated in batches, with each batch using an updated goal/step taxonomy. We stopped iterating when we finished annotating all the data. Any erroneous annotations were added back to the next batch to be re-annotated.
>
> Comment) ```L162: the taxonomy initialization is relying on the LLaMA model. How to make sure the model is not biased and is correct? How to prompt the model to give the right granularity level of steps, which is too detailed or too broad?```
>
> We want to emphasize that step candidates generated by LLaMA are merely used as options for annotators to choose from. They were free to add new step names as they wanted -- in fact, about 40% of times they added custom names. As a result, while some LLaMA suggestions may be biased or inaccurate, their optional use and the manual validation/correction by the annotators has reduced to a minimum any LLM-induced bias in the dataset.

---

> > ### Author Response · Authors · 2023-08-21
> > **(response 2/2)**
> >
> > Comment) ```L200: “This suggests that most of the step/substep segments capture higher-level actions than [22, 34] without sacrificing annotation density or capturing short-term atomic actions.” There is no clue for this conclusion in the context. More explanation is needed for getting this claim.```
> >
> > This was based on our evidence of longer temporal segments after comparing the step/sub-step categories to those in existing datasets. To avoid confusion, we have revised the text by replacing “higher-level actions” with “longer-duration actions”.
> >
> > Comment) ```Fig 4 middle: the figure is very interesting, in that performance decreases as the long-term memory length is close to 128. More analysis and discussion about this phenomenon are needed for this.```
> >
> > We believe that this is due to the large variance in our experimental results, as Fig 4-middle was plotted based on 3 runs. We plan to run more experiments to reduce the variance; we will update the paper when the results are ready.
> >
> > Comment) ```After reading the main paper and supplemental, the actual data annotation process is still not clearly stated. For example, how to select or train the annotators, how many iterations were needed, and when to do the many taxonomy updates.```
> >
> > The hiring, selection, and training of professional annotators were carried out by our third-party vendor, who specializes in data annotation. For quality control, they follow an established process: annotators are initially placed in the "training" queue to become familiar with the annotation task. Subsequently, they are moved to the "production" queue only if their annotations pass the quality test, which is evaluated by a separate team of auditors.
> >
> > We took 10 iterations for goal annotation, 10 iterations for step annotation, and 8 iterations for sub-step annotation. Each iteration involved annotators providing labels for a batch of videos, us reviewing them, and updating the taxonomy (meaning, the taxonomy update was done after each iteration). We have updated the text with these details in Section 3.3.

---

### Official Review · Reviewer_YdTM · 2023-07-24

**Rating:** 7
**Confidence:** 3
**Correctness:** The dataset construction is technical…
**Clarity:** Yes.

**Strengths:**

Strengths of the paper:

1. The motivation and methodology for constructing the dataset are both sound and interesting. It results in dense step annotations, addressing any potential label-data alignment issues.

2. The proposed dataset offers a large hierarchical taxonomy of goal-oriented activity labels for human activity recognition. Additionally, it provides valuable auxiliary information, such as natural language summary descriptions, step completion status, and step-to-goal relevance data. I believe that it is potentially valuable for the human activity recognition community, particularly for related tasks.

3. The paper is well-organized, providing sufficient details on data construction, baseline implementation, and evaluation.

Overall, the paper showcases good aspects in terms of its methodological approach, the significance of the proposed dataset, and the well-structured presentation.

**Additional Feedback:**

N/A

**Documentation:**

Yes, it is well-documented with sufficient detail.

**Limitations:**

The paper's limitations were only briefly acknowledged, with a mention of potential negative societal impacts, such as biases towards certain activities. To address this, a simple suggestion would be to expand the dataset collection and incorporate annotations for a more comprehensive representation of activities.

**Opportunities For Improvement:**

1. The introduction of each task is rather brief. A more comprehensive and formally stated problem for each task would enhance reader comprehension.

2. Only one baseline is presented for each task. Providing multiple baselines would be beneficial for future research and comparative analysis.

3. Although the supp. material includes training details of the model for each task, the description of the models themselves is insufficiently covered. This limitation may hinder readers' ability to grasp the intricacies of the baselines and the tasks.

**Relation To Prior Work:**

> Is it clearly discussed how this work differs from previous contributions?

Yes, it is. The authors also provide a good comparison with other works in Table 1.

**Summary And Contributions:**

The paper introduces "Ego4D Goal-Step," a novel hierarchical taxonomy of goal-oriented activity labels for human activity recognition. Unlike existing datasets that primarily focus on low-level actions, this dataset provides dense annotations for 47K procedural step segments (422 hours) and high-level goal annotations for 2,742 hours of Ego4D videos.

The main contribution of the paper is a large and hierarchical dataset, Ego4D Goal-Step, which is substantially larger in size compared to existing datasets and contains hierarchical action labels (goals - steps - substeps).

---

> ### Author Response · Authors · 2023-08-21
> **Thank you! We have updated the paper based on your suggestions.**
>
> Thank you for the encouraging comments and the constructive suggestions. We have updated our paper based on your suggestions (highlighted in blue). We summarize our changes below.
>
> Comment) ```The introduction of each task is rather brief. A more comprehensive and formally stated problem for each task would enhance reader comprehension.```
>
> We have expanded Section 4.1 with formal task definitions and background information. We are happy to add further details if needed. Please let us know!
>
> Comment) ```Only one baseline is presented for each task. Providing multiple baselines would be beneficial for future research and comparative analysis.```
>
> Thank you for the suggestion. We plan to incorporate an additional baseline method, EgoOnly [Wang et al., 2023] (https://arxiv.org/abs/2301.01380), for both the localization and online detection tasks. The model is currently achieving SoTA results on various egocentric activity detection benchmark, which is why we believe it would be a strong and interesting baseline to include.
>
> Comment) ```Although the supp. material includes training details of the model for each task, the description of the models themselves is insufficiently covered. This limitation may hinder readers' ability to grasp the intricacies of the baselines and the tasks.```
>
> We have added model descriptions to Appendix D in the supplementary material (highlighted in blue).

---

> > ### Comment · Reviewer_YdTM · 2023-08-30
> > **Official Comment of Reviewer YdTM**
> >
> > Thank you for your response which provided more detail to my questions and concerns.

---

### Author Response · Authors · 2023-06-16
**Downloading a private release of the dataset**

Dear reviewers,

As noted in the submission, we are doing a private release of the dataset. Please visit this link to access the data: https://people.csail.mit.edu/yalesong/pds/neurips23_ego4d_goalstep.zip

We kindly request that the reviewers refrain from redistributing this dataset until its official release. Thank you!

---

### Decision · Program_Chairs · 2023-09-22

**Decision:**

Accept (Spotlight)

**Comment:**

The paper has many merits as pointed out by the reviewers who gave accept ratings. Although there is still one review on the negative side, the concerns are largely addressed in the rebuttal.